# Italy without Urban 'Sprinkling'.
# A Uchronia for a Country that Needs a Retrofit of Its Urban and Landscape Planning

**Bernardino Romano \*, Lorena Fiorini and Alessandro Marucci**

Department of Civil and Environmental Engineering, University of L'Aquila, 67100 L'Aquila, Italy;
lorena.fiorini@univaq.it (L.F.); alessandro.marucci@univaq.it (A.M.)
\* Correspondence: bernardino.romano@univaq.it

**Abstract:** The research presented in the work is linked to important production of data over 10 years of activity that allowed us to trace the configuration of Italian urban settlements in the 1950s. Starting from this information, the paper puts forward a uchronian reconstruction of the physiognomy of the national territory asking if—instead of the weak urban development policies implemented for over half a century—a more purposeful method of planning and designing settlements had been chosen in the Sixties to favor their aggregation and protect the country's huge landscape heritage. From the model used, important indications emerge for control and management of retrofit (de-sprinkling) policies of which the need has been felt in recent years, as suggested by repeated messages from European bodies, the scientific community, associations and some politicians. The uchronic scenario is constructed starting from the settlement configuration of the 1950s, developing a model of maximum aggregation for the urbanized parts that were intervened in between this period and 2000, simulating a geography that maintains the quantities of soil transformed over the last 50 years. It emerges from the processing of the data that the Italian territory would have retained its low settlement density areas almost intact at the same level as in the 50s, that is to say 73% of the entire peninsular territory. It would also have preserved a large part of its free peninsular and insular coastline at about 60–70%, against the present day 45%.

**Keywords:** land use change; land take scenario; urban growth

## 1. Introduction

What would present-day Italy be like, if instead of implementing scarcely supervised procedures fostering fragmentary settlement growth, different decisions had been adopted in the Sixties favoring a form of territorial governance that was more mindful of public interests and not driven by ultra-powerful building sector enterprises? What would it be like in 30 years' time if, instead of continuing as it has, territorial governance took a different direction and territorial planning were strengthened significantly?

Numerous international studies and articles have shown in past years that Italian urban settlements follow one of the most dispersive models in Europe entailing extremely disparate and severe issues [1–4]. As is well known, many northern European Union (EU) countries are affected by extensive forms of sprawl [5–8], but in Italy's case, a different pattern has been described, called "sprinkling", much more finely dispersed and spontaneous, scarcely regulated by territorial planning tools and almost lacking in preventive urban planning [9]. This is the substantial difference: sprawl implies "patchy" growth, while sprinkling is "finely dispersed" growth. Built-up land (Figure 1), including the dense road network, covers almost 10% of the national territory in Italy, equal to the surface area of Belgium.

Of course, both the "sprawl" and the "sprinkling" are forms of low-density urban expansion on very extensive territorial surfaces, but sprinkling cannot be considered a particular case of urban sprawl, because the two structures have a different origin that depends on very differentiated approaches to urban planning, housing culture and building/territorial policy. In fact it is quite difficult to find mixed models in the same countries, at least as far as the settlement developed in the most recent decades is concerned: sprawl and sprinkling represent "signatures" of very clear settlement behavior in the contemporary urban landscape of the various countries, at least European. Regarding the issues and pathological aspects of these configurations of urbanized areas, it is possible to consult the numerous works published over the past decade that have also involved European and international institutions [10–14].

Even the events that caused the current Italian settlement model, starting from the Marshall Plan after World War II [15–17], have been described in detail by the authors of this paper in some other previous articles [18] to which reference is made for further details. This paper puts forward a uchronian reconstruction of the physiognomy of the national territory, if—instead of the weak urban development policies implemented for over half a century—a more purposeful method of planning and designing settlements had been chosen in the Sixties to favor their aggregation and protect the country's huge landscape heritage. This is not an exercise of inferential simulation of urban development, widely used in international literature for years [19–24], nor fantasy urban planning. The uchronia produced may be useful in understanding the severe degradation caused by over half a century of low territorial control. It can also serve as a starting point for the development of retrofit (de-sprinkling) policies of which the need has been felt in recent years, as suggested by repeated messages from the scientific community, associations and some politicians [25–27].

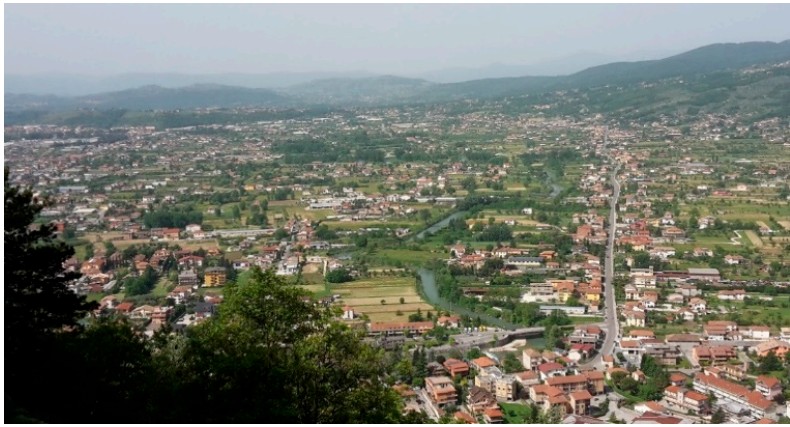

**Figure 1.** The characteristic pattern of Italian urban "sprinkling" in a flat area between the regions of Lazio and Abruzzo (central Italy).

## 2. Methods, Data, and Study Area

Why did we use a uchronian method? As is well known, this is a theoretical development hypothesis, differing from what has actually occurred. However it plots a course that would actually have been possible and, therefore, not merely imaginary. The main goal of the uchronia, in this case, consists in analytically proving, through the use of specific indicators, that an alternative development of the geographical distribution and density of Italian urbanized areas could have been achieved using an appropriate methodological approach and the necessary political ability. Proving the possible implementation of this alternative hypothesis and its advantages helps show that today it is still possible to work using a better method, whilst gaining awareness of the mistakes made according to the lesson-learning criterion. A critique that could be made of the uchronian hypothesis is that our present-day, more advanced cultural vision allows us to perceive issues differently and implement solutions using tools there were not available 50 years ago. However, in the case of settlement physiognomy this argument is not sound, since in the years in which Italian cities and

infrastructure grew at a fast pace, the same occurred in other northern European countries, but using more efficient town planning governance criteria and modalities that have led to far better results from numerous viewpoints.

We extended our study to the entire national territory using 3 × 3 km (9 km²) plots. On the basis of these plots, we calculated urbanization density in the 1950s and in the post-2000 period.

The data used for the chronological section of the 1950s come from the 1: 25.000 cartography by the Italian Military Geographical Institute (IGMI) between 1949 and 1962 and the methodological aspects have been repeatedly emphasized in other articles by the same authors [9,18], to which we refer for details. In these articles, however, the data on historical urbanization have been used only for the comparison with the current settlement situation to analyze and diagnose the evolutionary dynamics of urban growth, while the objective of the present work is to design an alternative model of settlement development starting from the same start condition of the 50s.

The results obtained from the study on 1:25,000 maps were then compared with those of urbanized areas available in vector format from regional maps generally derived from photo interpretation on a nominal scale of 1:10,000 or 1:5000. All Italian regions have produced land-use maps (LUMs) or regional technical maps covering the entire territory, almost always available on institutional geoportals. These maps allow you to select urbanized surfaces of various sorts. Although the level of accuracy of surveys is very high, there is an issue regarding the heterogeneity of data, both in terms of acquisition techniques and updating dates. The scale of detail is fairly standardized and generally of about 1:10.000, while the updating interval ranges between 2001 and 2012. Nine regions out of 20 have data dating back to around 2000, another nine updated their data around 2007, while only two (Liguria and Basilicata) have more recent data, ranging between 2012 and 2015. At first glance, a time interval over 10 years might seem too long to define the physiognomy of "post 2000" settlements uniformly. However, most changes in Italian urbanized areas occurred between the '60s and the '90s, while in the following years their growth slowed down considerably, especially in more saturated areas or areas having limited economic dynamics. In this regard, the comparison made using Corine Land Cover (CLC) data shows that between 1990 and 2000 urbanized areas grew by 10%, while between 1950 and 2000 they grew by 270% [28](Büttner et al., 2004).

More up-to-date and efficient data were produced from 2015 onwards by the National Environmental Research Institute [29].

For this study we chose to use regional LUM data and not the ISPRA data updated in 2017, since the latter also include surfaces covered by some categories of interurban roads (not separable) and, therefore, cannot be compared with the '50s data that did not include these elements. The urbanized areas surveyed by ISPRA [30] exceed those of regional LUMs by approximately 260,000 ha. This can be ascribed, in part, to the increase that has occurred over the past 10 years, but also largely to the inclusion of the road network amounting to almost 200,000 km out of the approximately 870,000 total in Italy comprising all categories (railway, highway, motorway, primary, secondary, tertiary, residential) extract by https://www.openstreetmap.org/.

This concept can be translated using the following equation (Figure 2):

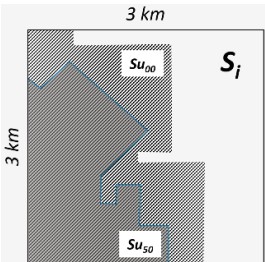

**Figure 2.** Diagram of the surfaces considered in 3 × 3 km plots.

$$\Delta S_{00\text{-}50} = \sum_{i=1}^{m}[(1 - d_i)\, S_i] \tag{1}$$

where:

$$\Delta S_{00\text{-}50} = Su_{00} - Su_{50} \tag{2}$$

$$d_i = \frac{Su_{50}}{Si} \tag{3}$$

$Si$ = *plot surfaces making up the analysis grid*

$Su_{00}$ = *Urban area Noughties*

$Su_{50}$ = *Urban area 50s*

$d_i$ = *Urban density of i-teenth plot*

where $m$ is the minimum number of plots which, with urban density in decreasing order, reaches the void density threshold (cut off density-$d_{co}$) equal to $\Delta S_{00\text{-}50}$ (Table 1).

We performed our simulation using the same demographic growth and the same urbanized surfaces, considering those listed in our Introduction. We included areas classified as such in the functional meaning of the term, like public parks, private gardens, squares and ancillary areas. The results obtained did not postulate vertical construction or volume increase scenarios, thus keeping the following parameters constant at regional level:

$$Su_{pc} = \frac{S_u}{n_{inhab}} \tag{4}$$

$$UD_{reg} = \frac{S_u}{S_{reg}} \tag{5}$$

$$Us = \frac{\Delta S_{00-50}}{\Delta t} \tag{6}$$

where:

$Su_{pc}$ = *Per capita urbanization (m²/inhab)*

$S_u$ = *Urbanized area (ha)*

$n_{inhab}$ = *Number of resident inhabitants*

$UD_{reg}$ = *Mean regional urban density (%)*

$S_{reg}$ = *Regional area (ha)*

$Us$ = *Mean urbanization rate (m²/day)*

$$\Delta t = 50 \; years \; (18.250 \; days)$$

## 3. Results

Between the '50s and the year 2000 Italian urbanized areas, defined as mentioned previously, increased by almost 1.5 million hectares. This is equivalent to a three-fold increase compared to the immediate post-war period, exceeding 7% of the entire national territory and reaching 10%, if we include all road surfaces too. However, some regions, such as Puglia and Tuscany, have witnessed five-fold increases, with profound changes in their landscape and territorial physiognomies. This has also been the case in Lombardy and Veneto, the Italian regions with the highest urban density, now at 14%. By applying function (1), we were able to develop a countrywide simulation of the extreme compaction of urbanized parts over the past half a century, maintaining their overall value (1.5 million hectares) as an invariant. Through the curves shown in Figure 3, we obtained the cutoff urban density ($d_{co}$) for every region, where $d_{co}$ is the urban limit density above which the uchronic model foresees the filling to receive all the urbanized surface realized from the 50s to 2000.

In other words, this threshold includes the minimum number of 9 km$^2$ plots that would have to be 100% saturated to contain all "new" urban areas in the region, thus avoiding the distribution of these transformed surfaces over large sections of farmland and semi-natural/natural areas. From this definition of the $d_{co}$ we are able to deduce that a low $d_{co}$ threshold is obtained when a limited urban increase ($\Delta S_{00\text{-}50}$) intervenes in a territory that previously had a high level of dispersion. However, at the same limited $\Delta S_{00\text{-}50}$ value, if it intervenes in cases of greater historical urban concentration, it corresponds to a high $d_{co}$ value, and in determining these conditions geo-morphologies, types of economy and, consequently, the historical development of the settlement in the various regions.

This classification emerges from Figure 3, in which various appear homogeneity in the latitudinal areas of the country: values above the national mean $d_{co}$ occur mainly in northern Italian regions and are higher in smaller and morphologically harsher territories (Valle d'Aosta, Friuli V.G. and Liguria), as well as in some central and southern regions, such as Marche, Umbria and Campania. There have been high urban concentration levels in interstitial valleys and lowlands in these areas since the '50s, while given its size Campania has witnessed a rather significant increase in $\Delta S_{00\text{-}50}$. The remaining central and southern regions have increasingly lower values that drop below 3% in mountainous or smaller regions (Abruzzo, Molise and Basilicata), or regions with vaster lowlands and relatively high settlement dispersion already in the '50s (Puglia, Tuscany and Emilia Romagna). All regions are far from the national mean $d_{co}$ value with a rather high standard deviation of 60% compared to the mean.

Based on these considerations, from Table 1 we may deduce that of the 11 Italian regions having a $d_{co} > 0.065$ (above the national mean, Figure 4) almost all, excluding Veneto and Lombardy, owe their high cutoff values to relatively limited urban growth in the time period considered. By contrast, of the nine regions with a $d_{co} < 0.065$, at least 4 (Puglia, Emilia Romagna, Tuscany and Lazio) owe their low cutoff to a relatively significant $\Delta S_{00\text{-}50}$ increase. At any rate, Figure 3 shows that $d_{co}$ values differ greatly from region to region: values range between 2% and 14%. Table 1 and Figure 5, highlight that the most significant correlations are tied to regional urban density and size of the same regions: the smaller the regions, the higher the threshold, which however rises slightly in the case of very vast regions. Therefore, we can confirm that $d_{co}$ depends mostly on mean regional urban density in the '50s, but also on size and morphology of regions, as well as urban growth in the time period considered.

This independence in land transformation is very evident even in the diagrams showing settlement scenarios tied to the uchronian simulation (Figure 6).

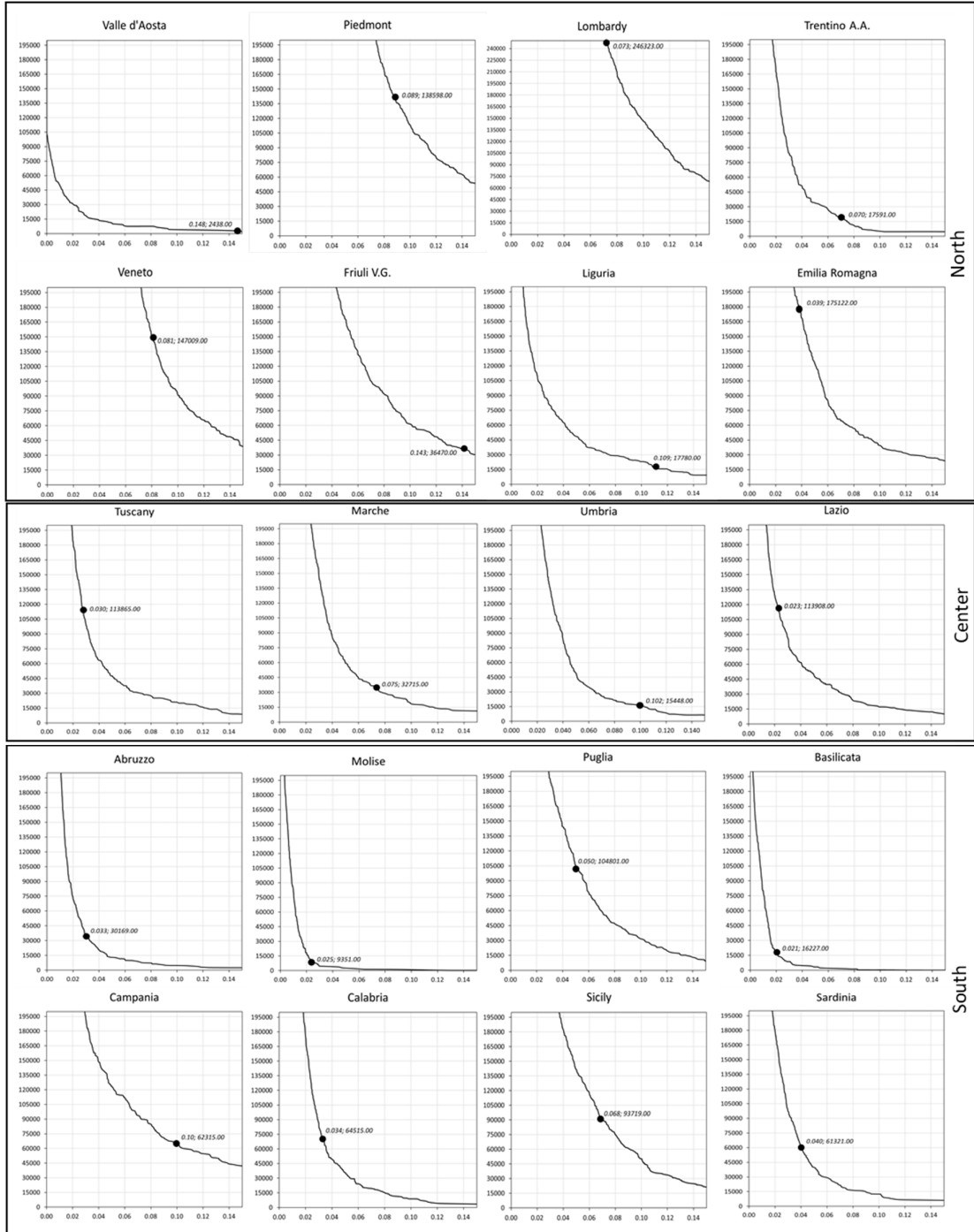

**Figure 3.** Curves for the determination of cutoff urban density, with the indication of the $d_{co}$ (*x*-axis) and $\Delta S_{00\text{-}50}$ (ha–*y*-axis) values in ha for each Italian region (only Lombardy has a different scale on the *y*-axis for reasons tied to its size).

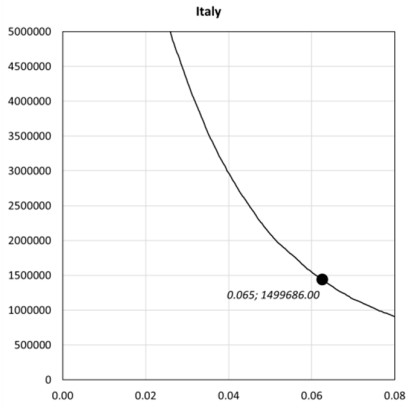

**Figure 4.** Mean $d_{co}$ value for Italy.

**Table 1.** Summary of the indicators used and their latitudinal breakdown in the country (N-North, C-Center, S-South).

| Region | Regional area (kmq) | Mean urban density 50s (%) | Increasing urban density 50s-2000 (%) | $\Delta S_{00-50}$ (ha) | dco (%) | Latitude | |
|---|---|---|---|---|---|---|---|
| Valle d'Aosta | 3260.92 | 0.070 | 0.007 | 2438 | 0.148 | N | |
| Friuli V.G. | 7859.92 | 0.043 | 0.046 | 36470 | 0.143 | N | |
| Liguria | 5405.9 | 0.024 | 0.033 | 17780 | 0.109 | N | |
| Umbria | 8461.07 | 0.019 | 0.018 | 15448 | 0.102 | C | |
| Campania | 13670.59 | 0.024 | 0.046 | 62315 | 0.100 | S | dco> 6,5% |
| Piedmont | 25387.07 | 0.035 | 0.055 | 138599 | 0.089 | N | |
| Veneto | 18415.46 | 0.037 | 0.080 | 147009 | 0.081 | N | |
| Marche | 9727.7 | 0.017 | 0.034 | 32715 | 0.075 | C | |
| Lombardy | 23863.86 | 0.040 | 0.103 | 246323 | 0.073 | N | |
| Trentino A.A. | 13604.72 | 0.008 | 0.013 | 17591 | 0.070 | N | |
| Sicily | 25832 | 0.013 | 0.036 | 93719 | 0.068 | S | |
| Puglia | 19533.85 | 0.011 | 0.054 | 104801 | 0.050 | S | |
| Sardinia | 24083.61 | 0.005 | 0.025 | 61321 | 0.040 | S | |
| Emilia-Romagna | 22123.24 | 0.015 | 0.079 | 175122 | 0.039 | N | |
| Calabria | 15221.61 | 0.009 | 0.042 | 64515 | 0.034 | S | |
| Abruzzo | 10826.99 | 0.007 | 0.028 | 30169 | 0.033 | S | dco < 6,5% |
| Tuscany | 22986.58 | 0.009 | 0.050 | 113866 | 0.030 | C | |
| Molise | 4461.03 | 0.005 | 0.021 | 9351 | 0.025 | S | |
| Lazio | 17206.403 | 0.015 | 0.066 | 113908 | 0.023 | C | |
| Basilicata | 9986.27 | 0.002 | 0.016 | 16227 | 0.021 | S | |
| ITALY | 301918.793 | 0.018 | 0.050 | 1499686 | 0.065 | | |

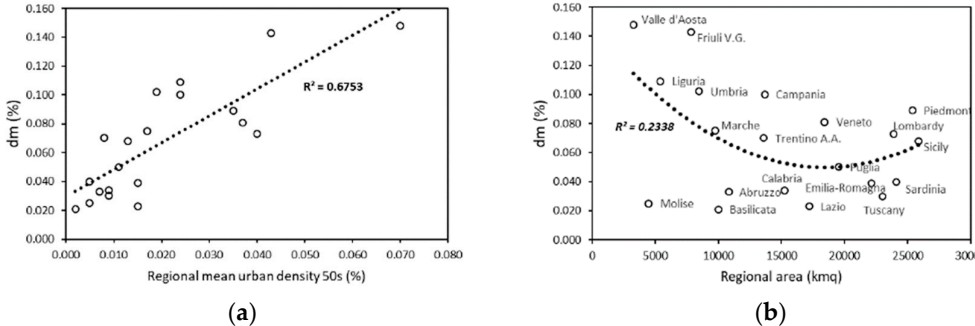

**Figure 5.** Correlation curves between $d_{co}$ and mean regional urban density in the 1950s (**a**) and regional territorial size (**b**).

The curves in Figure 6 were plotted by placing on the *x*-axis the six urban density (UD) ranges of the 3 × 3 km plots corresponding to the following urbanization threshold levels: irrelevant (<2%), rural (5%), peri-urban (10%), semi-saturated urban fabric (25%) and saturated urban fabric (50%). On the *y*-axis we have the percentage of plots in each region corresponding to the six different UD ranges. The three plotted curves show the situation in the immediate post-World War II period ('50s), in 2000 (Noughties) and in the hypothetical uchronian scenario of maximum compaction of urban growth. If we focus our attention on the curve of the first time period ('50s), we see significant differences in the three latitudinal areas of the country: North, Center and South. In the northern regions, the rate of virtually undisturbed plots (UD < 2%) was predominant and nearing 100% in territories having a mountainous morphology, such as Valle d'Aosta and Trentino A.A., while it was still high (>75%) in Liguria (this region too is very small and has an unfavorable morphology) and Emilia Romagna (a region having an almost entirely rural economy). The regions that already had industrial production sites and extremely dispersed settlement forms had far lower rates of low urban-density plots, amounting to 50% or less. Excluding Umbria, Marche and Campania, which are peninsular, hilly regions historically marked by significant settlement dispersion, all the other central and southern regions had extremely high values (still nearing 100%) in the territorial sections with a UD < 2%. Even the national mean (Figure 7) is in line with the mean of southern regions of around 75%. The situation changes radically after 2000, when this dispersive model spreads to all regions, leading to a significant drop in non-urbanized territorial sections (UD < 2%) that are halved almost everywhere. Only mountainous regions in the North (Valle d'Aosta and Trentino) and Basilicata in the South differ from this generalized phenomenon. The previously undisturbed parts of the territory are filled in part with buildings and urbanized surfaces that increase the intermediate density categories, i.e., those ranging between 5% and 25%, but, excluding Lombardy, never reach major saturation levels (UD > 25% and UD > 50%), underscoring the finely dispersed territorial distribution of settlements well represented by the "sprinkling" model. After 2000, less than 50% of the country (Figure 7) remains practically free from forms of land transformation for settlement purposes, but this is largely accounted for by vast mountain areas in the Alps and Apennines, in addition to some more intensive farming areas. Most of the coastal lowlands and inland valleys are covered by more or less dense/dispersed urban sprinkling, depending on mobility infrastructure configuration and morphology. As mentioned in our Introduction, the Italian configuration of urbanized areas in the last 50 years is extremely finely dispersed and is the product of well-known settlement growth processes. The model has been described using the Italian geographical sample and the major differences with sprawl highlighted. For many years, this term was used in national urban planning culture to define Italy's finely dispersed urban development. However, "sprinkling" seems to be more suitable to represent the configuration of Italy's urban constellations, present in other southern European countries too and in other continental areas, albeit with varying physiognomies, such as the Balkans, Greece and Portugal, or China and Japan. These examples may include various prevailing dispersion geometries, including the "linear" or the "urban dust", but more frequently they are mixed (as in Italy) with the distribution of tiny parts built on very large territories.

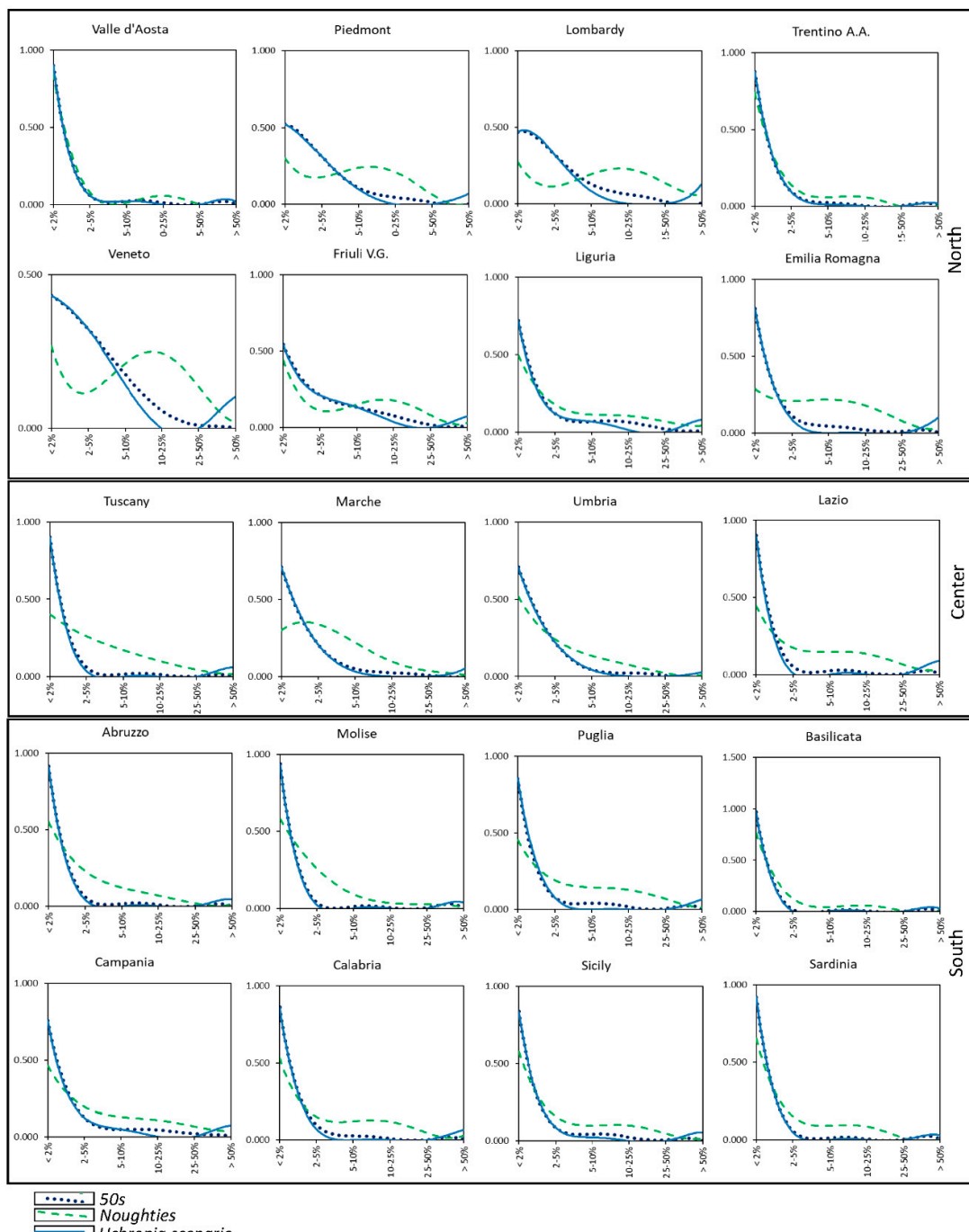

**Figure 6.** Curves showing variations in the six urban density categories assessed using 3 × 3 km plots in the '50s, the year 2000 and the uchronian scenario.

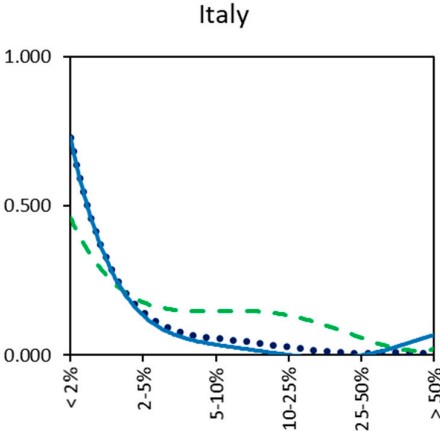

**Figure 7.** Figure 6 diagram for Italy.

If sprawl means "the spreading of urban developments (such as residential and commercial areas) over undeveloped land near cities", "sprinkling" expresses "scattered drop- or speck-like distribution". Sprinkling is a type of settlement that has already been experimentally classified using ad hoc indicators too [31–35] it.wikipedia.org/wiki/Sprinkling.

The initial third of the uchronia scenario curve overlaps with the one of the '50s in almost all regions, providing a picture of the conservation of lower UD (<5%) areas. In the hypothetical aggregation that this curve expresses, even intermediate-density plots (from 5 to 25%) decrease, while, of course, territorial sections saturated above 25% and 50%—absorbing all settlement growth between the '50s and 2000—increase. This is clearly an extreme simulation, the overall effects of which are shown in Figure 8, that would have undoubtedly required profound urban planning and design rigor. However, it would have helped preserve vast expanses of natural, semi-natural and rural landscape from irreversible and spatially pervasive degradation. The national landscape physiognomy would have been radically different, especially in some areas profoundly altered by soil sealing, such as lowlands and coastal strips.

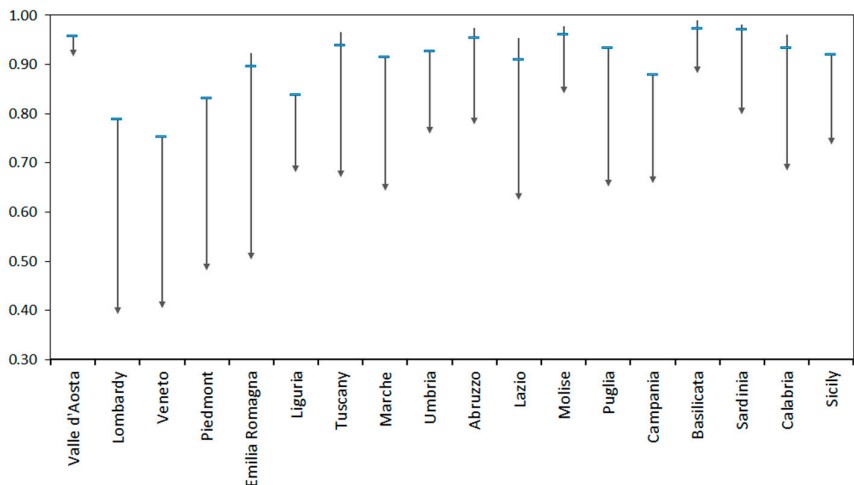

**Figure 8.** In this graph, with the regions arranged in North-South latitudinal order, the arrows show the contraction in regional rates of 3 × 3 plots having an urban density below 5% between the 1950s and 2000. The vertical segments show rate levels that would have been reached through the implementation of the uchronian scenario of aggregation, which are very close to or match post-war levels in all the regions.

Such a scenario would have required a robust urban plan to compact covered surfaces and appurtenances and produce an urban fabric that is more similar to the international standard of "sprawl", instead of the aforementioned "sprinkling" model described for Italy.

The results of the aggregation expressed by function [1] are shown in Figure 9 (for the entire peninsular territory) and Figure 10 for some regional samples. The maps have been drawn by graphically representing the urban density ranges $d_i$ in actual (9a) and uchronian (9b) conditions, while in the case of the samples in Figure 10 densities in the '50s, today and in the uchronian scenario are compared.

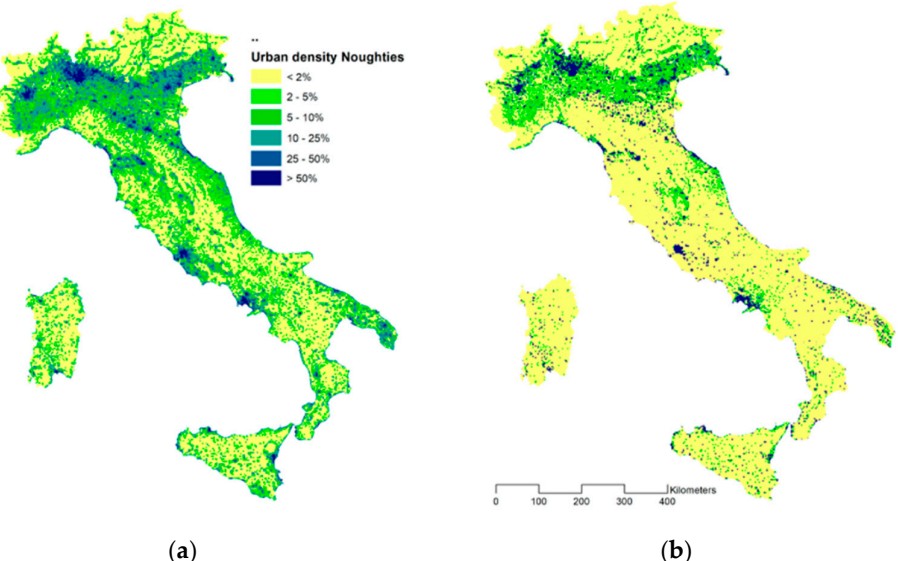

(**a**)　　　　　　　　　　　　　　　　　　　　　　　(**b**)

**Figure 9.** The configuration of the six urban density categories assessed on the basis of 3 × 3 km plots in 2000 (**a**) and in the uchronian scenario (**b**).

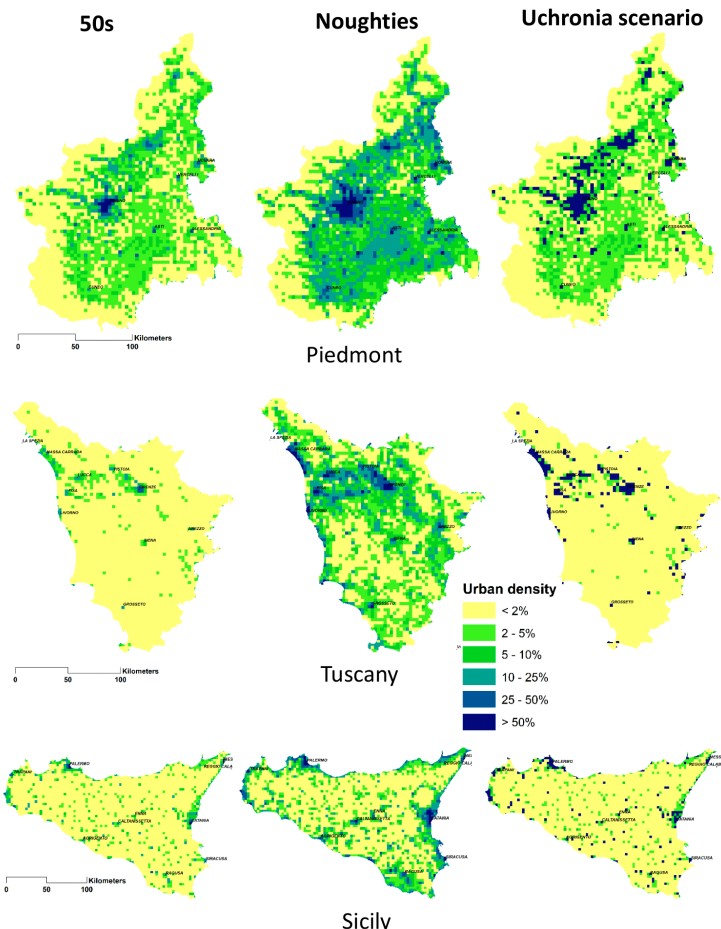

**Figure 10.** Some regional samples comparing distribution variation across the six urban density categories in the '50s, in 2000 and in the uchronian scenario.

In this scenario, the Italian territory (Figure 9a), would have preserved its low settlement density areas (UD < 5%) intact at the same level as in the '50s, that is to say 73% of the entire peninsular territory (Figure 9b). It would also have preserved most of its peninsular and insular-free coasts at about 60–70%, compared to the present-day just over 45% [36,37]. However, this model of extreme dispersion has led to the unrestrained road network growth with significant consequences for the fragmentation of the country's ecosystems that contain biodiversity of worldwide conservation interest.

Analyzing the effects on some categories of significant environmental, landscape and productive value, such as lowlands, hills and coastal strips, protected natural areas and habitats of European interest censused as Sites of Community Interest (SCIs) Natura 2000 (Figure 11), the implementation of the uchronian scenario would have left the rate of plots unaffected by settlements at the same level as in the '50s in all the territorial units considered, with peaks of 60% and even 70% of free areas. In reality, areas having a low-density dispersion (2–5%) have grown in the post-2000 years across all categories, excluding coastlines, where they have remained unchanged, and lowlands, where they dropped by 10% and have been transformed into higher density plots (5–10%). The uchronian scenario slightly reduces the number of low-density areas (2–5%) compared to the '50s, as it uses them in part for compaction, given that in various regions the $d_{co}$ has a cutoff ranging between these two values (Sardinia. Emilia, Calabria, Molise, Lazio, Abruzzo, Tuscany and Basilicata).

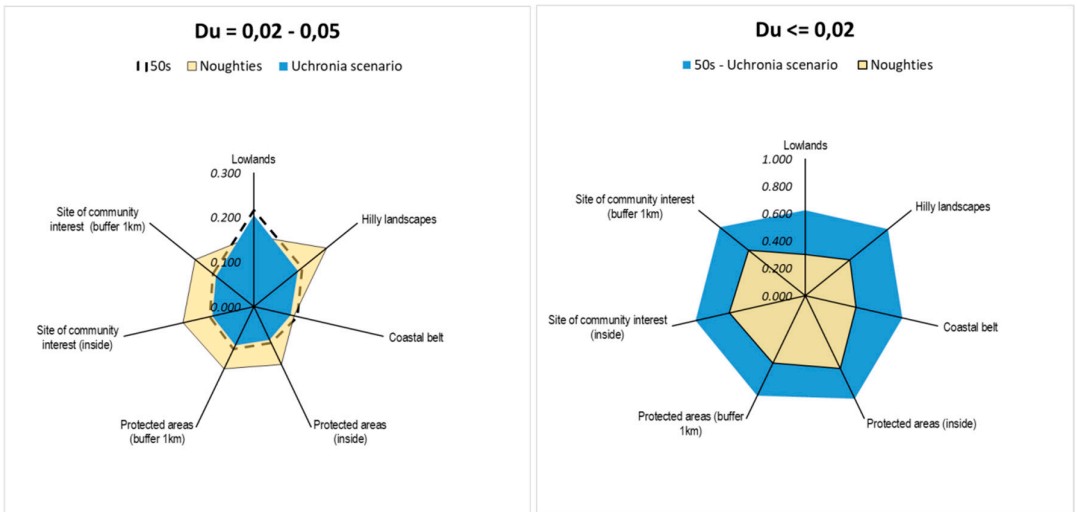

**Figure 11.** The rates of irrelevant (<2%) and low (2–5%) urban density areas corresponding to seven environmental and morphological units characterizing the country's landscape quality and perception in the following time periods: '50s, 2000 and the uchronian scenario.

## 4. Discussion

Is the scenario outlined in this paper thought-provoking? Yes, certainly. The hinterlands of main historical conurbations (as shown by regional examples in Figure 10) would have been saturated. This might suggest major degradation of important cultural and artistic heritage. However, most of what actually happened, in the absence of adequate urban planning, has produced severe aesthetical and functional alteration even in major historical city centers. Today, these are engulfed in the patchy and fragmentary matrices of very heterogeneous settlements in terms of type and functions. The real difference would not have been made by a different model of aggregation of surfaces or volumes, but by "plans" and "projects" that were drastically lacking in their more efficient forms.

Rural and natural landscapes of all sorts would have been less "under siege" by urbanization and the proliferation of roads. Indeed, one of the direct effects of urban aggregation is low infrastructure rate in territories, expressed as km of roads/km², with increased efficiency of public transportation and significant reductions in ecosystemic fragmentation of habitats. Several studies have shown [38] that Italian protected areas and habitats of community importance are currently surrounded by considerably high settlement densities, even though within them there has not been any significant increase.

The fabrics of numerous major Italian towns include broad green areas, often of high landscape and historical-monumental worth and the idea of actually replacing them with building clusters would clearly be absurd and unacceptable. However, these invaluable places worth preserving would be unaltered in the uchronian scenario, given our previously mentioned definition of "urbanized areas" referring to functional and not only physical aspects. All the same, a systematic, effective and advanced urban development plan would have prevented Italian settlements from literally exploding in surrounding rural hinterlands, i.e., that highly protected countryside in northern European cultures, which in Italy too is marked by very high-level landscape, ecological and functional physiognomies. What has been and is still lacking is urban and territorial design, as the subsequent phase to planning expressed through municipal urban development tools. Furthermore, over the years the latter have increasingly lost their regulatory strength and have not even succeeded in controlling the spatial balance of planned urban development. It is sufficient to consider that 20% of the peninsular territory, having 10 million inhabitants, is managed by means of urban development plans dated over 25 years ago. This is the case of 300 municipalities in the North (1,000,000 inhabitants), 215 municipalities in Central Italy (1,800,000 inhabitants) and as many as 920 municipalities in the South, with 6,700,000 inhabitants. This form of territorial planning can also be

defined "molecular, as it is managed and applied by municipalities having on average a size of 36 km², and in some very extreme cases 1 km². Furthermore, it lacks strategic frameworks for content mosaicking, as this had not been provided for by legislation in this field, falling under regional competence. Our remarks convey the idea of Italy's difficulty in implementing efficient forms of territorial planning and help clarify the reasons for the current situation. Furthermore, the anomalous dispersion of built/urbanized areas in every territory has been undoubtedly affected by illegal building, sadly known as an Italian peculiarity even among the international general public. However, the more severe effects of land degradation and uptake should be ascribed to the insufficient level of supervision of extremely weak executive planning. The data processed to develop the uchronian scenario described in this paper—where the country would have retained its urban expanse comparable to the size of the whole of Belgium—suggest that a different course might have been possible, even though it would not have had the extreme physiognomy described in this paper. Nevertheless, from the more recent Sentinel high-resolution remote-sensing datasets processed by ISPRA, we know that the Italian territorial governance bodies, both at the national and local level, have not learned the lesson of the past 50 years yet. Indeed, in recent years, the phenomenon of land uptake persists with the same dispersive criteria used between the post-World War II period and the year 2000, although mitigated in terms of absolute magnitude from one region to the other.

## 5. Conclusions

The present-day economic conditions of the country and foreseeable conditions in the mid and long term clearly do not suggest the resumption of uncontrolled urbanization of the territory, comparable to what occurred in the second half of the 20th century. However, the studies conducted for over a decade have shown that compared to the "amount" of newly urbanized areas, what is more important is "how" this urbanization is distributed and, therefore, designed and planned. So, in the future, it will be very important for the country to adopt settlement growth models to compact and aggregate built-up and transformed parts, and carefully avoid further dispersion in all its forms.

This is not an easy task, since domestic politicians are still unprepared to tackle issues of this kind and tools such as "building amnesties"—the most recent of which is dated 2018—are still preferred.

The responsibilities of the critical national condition of urban settlement can be attributed to all administrative levels of the country. After 1942, the state no longer legislated on the transformation of the soil; the regions have produced many laws, up to recent years, but increasingly weakening strategic control over the activities of the municipalities; the latter have, therefore, accentuated their decision-making powers on the territories and favored a hypertrophy of urban expansions to increase social consent and also the tax revenues connected to the construction of new buildings. Today, even state bodies, such as the Ministry of the Environment or national research institutes, harshly criticize local policies of disorganized urban growth, but no one manages to reorganize the subject to achieve greater effectiveness and sustainability, also by reversing the trend.

On an international scale, urban compaction already offers many study experiences in the field of "de-sprawling" [39–44] which are beginning to find their way into urban development tools in the various countries that have suffered this phenomenon massively over the past 50 years. However, Italian dispersion, defined as "sprinkling", is very different from the international model of sprawl, both in terms of origin, land/property ownership regimes and functional types.

The calculations carried out and the indicators of the uchronian scenario show that it would still be possible to further curb land consumption if territorial policies to this end were implemented.

In fact, the results that emerge from the proposed uchronic scenario show that an alternative model to extreme dispersion is possible, provided that urban and strategic planning, with "horizon scanning" [45] criteria, is able to exercise control over some fundamental parameters: these include the cover ratio, the building density and the urbanization rate, introducing in municipal plans devices for coupling between the future urban dimensions and the real demographic dynamics. This last aspect also constitutes one of the requests expressed on the European scale in terms of land consumption [29].

For this purpose, as a future development, it could be interesting to assess the density of urbanized areas in protected and hydrogeological risk areas, in order to understand how much of this urbanization has taken place in compliance with regulations.

This would be a further "uchronian scenario in accordance with regulations" which, together with previous ones, could be of support to territorial planning with a view to curbing soil consumption.

For further details regarding the features of the sprinkling model, in our paper there are many references to already published scientific literature and some examples of methods on possible de-sprinkling procedures have already been produced [26]. However, this is a fertile area of research that deserves to be developed, as it will surely be a key aspect to be tackled in Italian territorial policy in decades to come.

**Author Contributions:** Conceptualization, B.R., L.F., A.M., Data curation, L.F. Methodology, B.R., Software, A.M., Supervision, B.R., Writing original draft, B.R.

**Funding:** This research was funded by Ministry of Education, University and Research, Basic Research 2017, items 295-302, Low 11.12.2016, n. 232

**Acknowledgments:** The methodology presented has been implemented in the research project and monitoring supported by Umbria Region. We are grateful to Francesco Zullo, WWF Italia Onlus and Cheryl Di Lorenzo for their collaboration.

**Conflicts of Interest:** The authors declare no conflict of interest.

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
