# Peer review of "Italy without Urban ‘Sprinkling’. A Uchronia for a Country that Needs a Retrofit of Its Urban and Landscape Planning"

_sustainability, doi:10.3390/su11123469_

Reviewer 1 Report

The aim is described in the Introduction to the text: "This paper puts forward a uchronian reconstruction of the physiognomy of the national territory,...". This aim could be considered relevant, but the manuscript does not clearly describe the innovation offered by this text compared to other previous works by the authors.

The following comments and suggestions are specifically made:

(1) The text presents notable similarities with the work of the authors cited as Romano et al., 2017b. This work is cited in line 62, but such connections and similarities are omitted. In this respect, literal fragments of the above mentioned text and other texts of the authors have been found between lines 46 to 86 and lines 119 to 146, which could be considered, according to this reviewer, an evident self-plagiarism of fragments of the original, not being considered in any case admissible.  It is therefore necessary, on the one hand, to highlight the connections of this original with previous works, beyond its mere quotation, incorporating mentions in other sections such as methodology, results, discussion and conclusions. The innovation of the text should be clearly and categorically evidenced as opposed to previous researches. On the other hand, any nonriginal fragment of the text should be suppressed.
(2) The abstract contains details that do not appear to be singularly relevant to this section, but does not provide details of the methodology carried out in the work.
(3) It is recommended to elaborate further on how the "indicators of the uchronian scenario" can be useful for future urban and regional planning.

Author Response

The aim is described in the Introduction to the text: "This paper puts forward a uchronian reconstruction of the physiognomy of the national territory,...". This aim could be considered relevant, but the manuscript does not clearly describe the innovation offered by this text compared to other previous works by the authors.

(1) The text presents notable similarities with the work of the authors cited as Romano et al., 2017b. This work is cited in line 62, but such connections and similarities are omitted. In this respect, literal fragments of the above mentioned text and other texts of the authors have been found between lines 46 to 86 and lines 119 to 146, which could be considered, according to this reviewer, an evident self-plagiarism of fragments of the original, not being considered in any case admissible.  It is therefore necessary, on the one hand, to highlight the connections of this original with previous works, beyond its mere quotation, incorporating mentions in other sections such as methodology, results, discussion and conclusions. The innovation of the text should be clearly and categorically evidenced as opposed to previous researches.

Are inserted the following text:

Even the events that caused the current Italian settlement model starting from the Marshall Plan after World War II (Milward, 1989; Bradford and Eichengreen, 1991; Raftopoulos, 2009) have been described in detail by the authors of this paper in some other articles previous (Romano et al., 2017b) to which reference is made for further details

 The data used for the chronological section of the 1950s come from the 1: 25.000 cartography by the Italian Military Geographical Institute (IGMI) between 1949 and 1962 and the methodological aspects have been repeatedly emphasized in other articles by the same authors (………… … ..), to which we refer for details. In these articles, however, the data on historical urbanization have been used only for the comparison with the current settlement situation to analyze and diagnose the evolutionary dynamics of urban growth, while the objective of the present work is to design an alternative model of settlement development starting from the same start condition of the 50s.

 On the other hand, any nonoriginal fragment of the text should be suppressed.

The parts already published in other works to which the previous statements refer, have been deleted.

(2) The abstract contains details that do not appear to be singularly relevant to this section, but does not provide details of the methodology carried out in the work.
The abstract has been totally reorganized eliminando la parte finale e inserendo un testo riferito alla metodologia.

The research presented in the work is linked to an important production of data in over ten years of activity and that allowed to trace the configuration of the Italian urban settlement in the 1950s. Starting from this information, the paper puts forward a uchronian reconstruction of the physiognomy of the national territory if - instead of the weak urban development policies implemented for over half a century - a more purposeful method of planning and designing settlements had been chosen in the Sixties to favor their aggregation and protect the country’s huge landscape heritage. From the model used, important indications emerge for control and management of retrofit (de-sprinkling) policies of which the need has been felt in recent years, as suggested by repeated messages from the European bodies, scientific community, associations and some politicians. The uchronic scenario is constructed starting from the settlement configuration of the 1950s, developing a model of maximum aggregation for the urbanized parts intervened between this chronology and 2000, simulating a geography that maintains the quantities of soil transformed over the last 50 years. It emerges from the processing of the data the Italian territory would have retained its low settlement density areas almost intact at the same level as in the 50s, that is to say 73% of the entire peninsular territory. It would also have preserved a large part of its free peninsular and insular coastline at about 60-70%, against the present day 45%.

 (3) It is recommended to elaborate further on how the "indicators of the uchronian scenario" can be useful for future urban and regional planning.

In fact, the results that emerge from the proposed uchronic scenario show that an alternative model to extreme dispersion is possible, provided that urban and strategic planning, with “horizon scanning” (………… ..) criteria, is able to exercise control over some fundamental parameters: these include the cover ratio, the building density and the urbanization rate, introducing in the municipal plans devices for coupling between the future urban dimensions and the real demographic dynamics. This last aspect also constitutes one of the requests expressed on the European scale in terms of land consumption.

Reviewer 2 Report

Comments to the Authors

The paper titled “Italy without urban “sprinkling”. A Uchronia for a country that needs the retrofit of its urban and landscape planning” presents an original point of view on the analysis and critique of the urban development of Italian peripheries.

The paper is well organized, but the abstract needs to be completely rewritten, at the moment is a mere repetition of the body text. Rows 10-13 are equal to Rows 88-91, Rows 14-16 are equal to Rows 94-97. Rows 16-19 are almost equal to Rows 322-325. Rows 21-24 are equal to Rows 363-365.

The use of the Uchronian method for sure gives to the text an interesting perspective about the history and the potential growth of the Italian periphery. It is not clear, nevertheless, if the proposed method of the “sprinkling” is a totally-Italian pattern or can be found on the broader scale of Europe or World, in Rows 298-303 there is a reference to – for instance – Greece or Portugal, but are not well described the “varying physiognomies”.

The literature review shows some inconsistencies: 9 up to 68 references (more than the 13%) refer to at least one of the authors and the main Wikipedia page, quoted as a reference has been written by one of the authors, this led to the risk of a high self-referencing of the entire paper.

Apart from that the paper is coherently organized and shows good originality, despite some minor part that should be better explored.

In Row 214 the definition of “cutoff urban density” is unclear and should be described more rigorously and with the proper emphasis.

 In Figure 6, the line of “50s” (light grey, dotted) is almost invisible both in pdf and in the printed copy.

I suggest visualizing graphically the Uchronian scenario, in this way a better understanding of the pattern, in opposition to the sprinkling pattern, will be reached.

In the section “Results”, in Rows 228-238 the authors stated a substantial homogeneity of the results, describing the differences: an analysis of the causes of those differences is required, also on a hypothetic basis.

Personally, and in this sense can be useful a paragraph of explanations, I am not sure if sprinkling can be interpreted as a particular case of the Urban Sprawl.

Finally, in the conclusions, my advice is to add a judgement on the land-use planning policies in Italy, if on one hand is clear the critique to the application of Law 1140/42,  on the other hand should be also clear the planning hypertrophy of standards, caused (according for instance to Avarello) by the overwhelming and often contradictory role of State and Regions in the development of planning policies.

Discussion, Conclusions and Bibliography are appropriated and balanced. Overall, the paper is clear and easy to understand. The results are interesting with good potential for future researches.

Hope my comments will be useful to the authors.

Author Response

The paper is well organized, but the abstract needs to be completely rewritten, at the moment is a mere repetition of the body text. Rows 10-13 are equal to Rows 88-91, Rows 14-16 are equal to Rows 94-97. Rows 16-19 are almost equal to Rows 322-325. Rows 21-24 are equal to Rows 363-365.

The abstract has been reorganized as follow:

 The research presented in the work is linked to an important production of data in over ten years of activity and that allowed to trace the configuration of the Italian urban settlement in the 1950s. Starting from this information, the paper puts forward a uchronian reconstruction of the physiognomy of the national territory if - instead of the weak urban development policies implemented for over half a century - a more purposeful method of planning and designing settlements had been chosen in the Sixties to favor their aggregation and protect the country’s huge landscape heritage. From the model used, important indications emerge for control and management of retrofit (de-sprinkling) policies of which the need has been felt in recent years, as suggested by repeated messages from the European bodies, scientific community, associations and some politicians. The uchronic scenario is constructed starting from the settlement configuration of the 1950s, developing a model of maximum aggregation for the urbanized parts intervened between this chronology and 2000, simulating a geography that maintains the quantities of soil transformed over the last 50 years. It emerges from the processing of the data the Italian territory would have retained its low settlement density areas almost intact at the same level as in the 50s, that is to say 73% of the entire peninsular territory. It would also have preserved a large part of its free peninsular and insular coastline at about 60-70%, against the present day 45%.

The use of the Uchronian method for sure gives to the text an interesting perspective about the history and the potential growth of the Italian periphery. It is not clear, nevertheless, if the proposed method of the “sprinkling” is a totally-Italian pattern or can be found on the broader scale of Europe or World, in Rows 298-303 there is a reference to – for instance – Greece or Portugal, but are not well described the “varying physiognomies”.

The text has been integrated as following:

The model has been described using the Italian geographical sample and the major differences with sprawl highlighted. For many years, this term was used in national urban planning culture to define Italian finely dispersed urban development. However, “sprinkling” seems to be more suitable to represent the configuration of Italy’s urban constellations, present in other southern European countries too and in other continental areas, albeit with varying physiognomies, such as the Balkans, Greece and  Portugal, or China and Japan. These examples may include various prevailing dispersion geometries, including the "linear" or the "urban dust", but more frequently they are mixed (as in Italy) with the distribution of tiny parts built on very large territories.

 The literature review shows some inconsistencies: 9 up to 68 references (more than the 13%) refer to at least one of the authors and the main Wikipedia page, quoted as a reference has been written by one of the authors, this led to the risk of a high self-referencing of the entire paper.

Three citations from the authors have been deleted.

Apart from that the paper is coherently organized and shows good originality, despite some minor part that should be better explored. In Row 214 the definition of “cutoff urban density” is unclear and should be described more rigorously and with the proper emphasis.

Has been inserted the following definition:

 Through the curves shown in Figure 3, we obtained the cutoff urban density (dco) for every region, where dco is the urban limit density above which the uchronic model foresees the filling to receive all the urbanized surface realized from the 50s to 2000.

In other words, this threshold includes…………..

 In Figure 6, the line of “50s” (light grey, dotted) is almost invisible both in pdf and in the printed copy.

Figure 6 has been redesigned by changing colors

I suggest visualizing graphically the Uchronian scenario, in this way a better understanding of the pattern, in opposition to the sprinkling pattern, will be reached.

Unfortunately I was not able to understand this observation, as the Ukrainian scenarios are graphically displayed in figures 9a and 9b and then in regional detail in Figure 10. If the reviewer can kindly better specify the request we can insert it.

In the section “Results”, in Rows 228-238 the authors stated a substantial homogeneity of the results, describing the differences: an analysis of the causes of those differences is required, also on a hypothetic basis.

The section has been reorganized as following

From this definition of the dco we are able to deduce that a low dco threshold is obtained when a limited urban increase (ΔS00-50) intervenes in a territory that previously had a high level of dispersion. However at the same limited ΔS00-50 value, if it intervenes in cases of greater historical urban concentration, corresponds to a high dco value, and in determining these conditions geo-morphologies, types of economy and, consequently, the historical development of the settlement in the various regions.  

This classification emerges from Figure 3, in which various appear homogeneity in the latitudinal areas of the country: values above the national mean dco occur mainly in northern Italian regions and are higher in smaller and morphologically harsher territories (Valle d’Aosta, Friuli V.G. and Liguria), as well as in some central and southern regions, such as Marche, Umbria and Campania. There have been high urban concentration levels in interstitial valleys and lowlands in these areas since the ‘50’s, while given its size Campania has witnessed a rather significant increase in ΔS00-50. The remaining central and southern regions have increasingly lower values that drop below 3% in mountainous or smaller regions (Abruzzo, Molise and Basilicata), or regions with vaster lowlands and relatively high settlement dispersion already in the ’50s (Puglia, Tuscany and Emilia Romagna)

Personally, and in this sense can be useful a paragraph of explanations, I am not sure if sprinkling can be interpreted as a particular case of the Urban Sprawl.

Has been inserted the following text

 Of course, both the "sprawl" and the "sprinkling" are forms of low density urban expansion on very extensive territorial surfaces, but sprinkling cannot be considered a particular case of urban sprawl, because the two structures have a different origin that depends on from very differentiated approaches to urban planning, housing culture and building/territorial policy. In fact it is quite difficult to find mixed models in the same countries, at least as far as the settlement developed in the most recent decades is concerned: sprawl and sprinkling represent "signatures" of very clear settlement behavior in the contemporary urban landscape of the various countries, at least European.

Finally, in the conclusions, my advice is to add a judgement on the land-use planning policies in Italy, if on one hand is clear the critique to the application of Law 1140/42,  on the other hand should be also clear the planning hypertrophy of standards, caused (according for instance to Avarello) by the overwhelming and often contradictory role of State and Regions in the development of planning policies.

Has been add the following text:

The responsibilities of the critical national condition of urban settlement can be attributed to all administrative levels of the country. After 1942 the State no longer legislated on the transformation of the soil; the Regions have produced many laws, up to recent years, but increasingly weakening strategic control over the activities of the municipalities; the latter have therefore accentuated their decision-making powers on the territories and favored a hypertrophy of urban expansions to increase social consents and also the tax revenues connected to the construction of new buildings. Today, even state bodies, such as the Ministry of the Environment or national research institutes, harshly criticize local policies of disorganized urban growth, but no one manages to reorganize the subject to achieve greater effectiveness and sustainability, also by reversing the trend.

 Round  2

Reviewer 1 Report

It is considered that the requested changes have been complied with.